# Meditation Effects on Anxiety and Resilience of Preadolescents and Adolescents: A Randomized Controlled Study

**DOI:** 10.3390/children8080689

**Published:** 2021-08-11

**Authors:** Alexandra Gomes, Joana Vieira dos Santos, Luís Sérgio Vieira

**Affiliations:** 1Psychology Research Centre (CIP-UAL), Universidade do Algarve, Gambelas, 8005-139 Faro, Portugal; asgomes@ualg.pt (A.G.); jcsantos@ualg.pt (J.V.d.S.); 2Department of Psychology and Educational Sciences, Universidade do Algarve, Gambelas, 8005-139 Faro, Portugal

**Keywords:** transcendental meditation, meditation, anxiety, resilience, experimental design

## Abstract

Meditation has been described as having a positive impact on well-being while reducing anxiety and stress among those who practice, mainly working as a resource to cope with everyday difficulties. As a simple and easy to apply meditation technique, transcendental meditation (TM) has shown promising results in adults and in children, although more studies are needed to show the impact on psychological and behavioral dimensions in children and adolescents. This quasi-experimental, pre-test–post-test study, with a control group, aimed to evaluate the impact of TM on the stress and resilience of children and adolescents, with ages between 9 and 16 years old. Participants were selected within schools which implemented the Quiet Time Program (QT), from those who volunteered to participate. They were randomly assigned to an experimental group (immediate TM learning) and to a control group (delayed TM learning). A repeated measures ANOVA showed an interaction of time and group on externalizing behavior, from the strengths and difficulties measure. The experimental group decreased on externalizing less adjusted behaviors, while the control group increased in this aspect, after a twelve-week period. TM failed to reduce anxiety and to contribute to resilience in the TM experimental group. Both groups improved anxiety indicators. The results might suggest students were acting upon their expectation of improvement on practicing TM or solely modifying their behavior along the contextual factors, which affected both groups equally.

## 1. Introduction

In general, the term “meditation” refers to mental and emotional control practices from many cultural contexts including those of Christianity and Islam, the most frequently applied being those originating from the Eastern spiritual traditions of India, Tibet, China, and Japan [1]. Meditation has been adopted in Western countries both as a spiritual practice and a mind–body therapeutic intervention [2].

Matko and Sedlmeier [3] stated that meditation is an umbrella term which includes a huge number of diverse practices. Meditation is often used to describe both the mental training technique employed by meditators and the state of consciousness resulting from it [4]. Recently, Ganguly et al. [5] considered a specific consciousness state in which deep relaxation and increased internalized attention co-exist.

Transcendental meditation (TM) is one of these meditation techniques. Transcendental meditation was brought to the West by Maharishi Mahesh Yogi from India. This type of meditation became more popular in 1960, and scientific research began to grow from 1970. This is a simple and natural method that allows the person to establish a peaceful environment for mental awareness. The central objective is to keep the person less involved as possible in physical and commonplace concerns, transferring this concentration to something bigger: their own position within a vast universe [6]. According to the literature review, transcendental meditation has numerous positive effects, such as: reduction in anxiety and stress, increased self-realization, increased creativity and concentration, increased autonomy and stability, decreased cortisol and noradrenaline, increased serotonin and dopamine, blood pressure control, and increased recall (e.g., [7]).

Several studies have been conducted demonstrating the Quiet Time Program/transcendental meditation (QT/TM) as a technique able to reduce psychological distress and improve cognitive and emotional functioning, resulting in less anxiety, more resilience, and better academic achievements. Elder et al.’s [8] study results indicated a reduction in psychological distress and anxiety among minority students practicing TM. The authors emphasized the importance of these results, due to the known association between psychological distress and poor school performance, and impaired physical and mental health. Colbert [9] suggested that TM can be an effective program for increasing graduation rates in urban schools. Wendt et al. [10], using TM with high school students, verified a significant reduction in anxiety and an increase in resilience. Additionally, time spent meditating correlated positively with resilience. Students participating in TM reported sleep improvements, more happiness, and higher self-confidence.

### 1.1. Anxiety

Polanczyk et al. [11], in a meta-analysis of the worldwide prevalence of mental disorders in children and adolescents, estimated that 13.4% of children and adolescents (6–18 years of age) have some form of mental disorder. Anxiety disorders showed the highest prevalence rate (6.5%). With this in mind, it is imperative to study the factors that contribute to the initiation, maintenance, or increase in anxiety, in order to fasten the detection of pathological anxiety and develop efficient treatments [12].

Anxiety, as a “normal” emotional reaction of the subject, adaptive and important to their survival, is defined as an unpleasant feeling that is associated with a feeling of anticipation of a future threat leading to muscular tension and states of vigilance. This is characterized by a set of physiological, behavioral, and cognitive changes that allow the subject to defend themself against possible threats to their integrity [13]. However, it can intensified, become pathological, and constitute a permanent form of reaction, reflecting in maladjustment and dysfunctionality when it produces disproportionate responses to reality, compromising the subject’s routine and activities [14].

Therefore, in this sense, Hartmann [15] distinguished between normal and pathological anxiety. Normal anxiety appears when someone is in a naturally threatening or scary situation, where anxiety is a normal and understandable reaction. The diagnosis of an anxiety disorder is associated with situations when an individual, in a normal daily situation, reacts with higher levels of anxiety or anticipates harm in an illogical way, where one could consider the diagnosis of an anxiety disorder/pathological anxiety.

In children and adolescents, anxiety disorders are a very common form of psychopathology and usually result in impairments in academic performance, socialization, and family functioning [16]. On the contrary, “normal” or “healthy” anxiety can be seen in a positive way, as it can help individuals in optimizing their performances and overcoming obstacles in their daily life [17]. For example, the previous author concluded that the right “amount” of anxiety helps company leaders to perform at their best level, build better teams, and improve productivity and morale.

### 1.2. Resilience

Grissom [18] stated that the key concern for those who study resilience is to better understand how students build the capacity to manage their challenges. Resilience can be defined as good outcomes despite the threats to adaptation or development [19].

According to many empirical studies, resilience is negatively correlated with indicators of mental ill-being, such as depression, anxiety, and negative emotions [20]. Abiola and Udofia [21] suggested that resilience is associated with increased quality of life, well-being, and functional capacity in times of adversity. Having a resilience streak allows overcoming stress or problems and becoming stronger with daily experiences [22]. Resilience is suggested as a buffer between burnout and adverse physical and mental health outcomes [23]. Improving the levels of resilience may be one way of preventing adolescent anxiety.

This randomized controlled study aimed to investigate the effects of meditation with students on both anxiety and resilience. We hypothesized that the experimental group would present less anxiety and higher resilience after a twelve-week period of practicing TM, when compared to a similar control group (no intervention group).

## 2. Materials and Methods

This was a quasi-experimental study design, using a pre-test–post-test methodology, with a control group.

### 2.1. Participants

The participants were recruited from a school cluster from the Portuguese public educational system, the pilot in implementing the Quiet Time Program. The implementation team pitched to a total of 10 school clusters, but a school cluster in Algarve was selected to be the pilot for the Quiet Time Program. Other school clusters were included in the program, but the data refer to the pilot school cluster.

A full presentation of the program was presented to the school community: teachers and staff, students, and parents. An informed consent form was delivered to all parents or legal equivalents. Teachers and staff gave their informed consent to participate and were also informed about the TM technique.

From the 300 consents received, 200 students were randomly selected to participate, but only 168 completed the full experimental design. The class where children pertained to the study design were randomly assigned to experimental and control groups. A total of 102 participants were integrated into the experimental group, which learned and practiced transcendental meditation in school. A total of 66 participants were integrated into the control group.

All students who showed interest were included. Exclusion criteria related only to an inability to understand or implement the TM technique. Hence, students with severe cognitive impairments were not included in the sample.

The final sample consisted of 168 students from the 2nd and 3rd educational cycles: 5th grade (13.10%), 6th grade (21.43%), 7th grade (32.4%), 8th grade (18.45%), and 9th grade (14.29). The students presented a retention frequency of 17.68% (1 out of 5 had at least a retention in their school path).

Participants were 12 years old on average (*M* = 12.46, *SD* = 1.548). The younger students were 9 years old, and the oldest were 16 years old. There were more girls (58.92%) than boys (38.10%) in the sample. A total of 2.9% did not identify as boy or girl for unknown reasons.

There were no differences at baseline considering age (*p = 0*.714), sex (*p* = *0*.864), and school year (*p* = *0*.406). Having a retention is associated with the experimental group (χ^2^ = 4.947, *p* = *0*.026).

### 2.2. Measurements

To measure the psychological dimensions in this study, we used three instruments: The Multidimensional Anxiety Scale for Children (MASC) [24]; the Strengths and Difficulties Questionnaire (SDQ) [25]; and the Health Kids Resilience Assessment Module of the California Healthy Kids Survey (CHKS) [26]. The first two measures were used to assess anxiety and the internal and external behavioral expressions of less adjusted mental and emotional states. The third measure concerned the operationalization of resilience.

#### 2.2.1. Multidimensional Anxiety Scale for Children (MASC)

The MASC is a normed self-report questionnaire that assesses anxiety in children and adolescents, adapted for Portuguese adolescents [24,27]. It uses four basic factors (physical symptoms, harm avoidance, social anxiety, and separation/panic), emerging from 39 items answered on a 4-point Likert scale. It is a scale with a reasonable to good internal consistency for all items and factors (between 0.74 and 0.90), strong convergent validity, and good reliability in the test–retest [27,28].

#### 2.2.2. Strengths and Difficulties Questionnaire (SDQ)

To measure the strengths and difficulties of the participants, we used the SDQ [29] adapted by Pechorro et al. [30] for Portugal. The SDQ is a preadolescent/adolescent brief behavior screening questionnaire. It uses 25 items describing psychological attributes, organized in the following components: emotional symptoms; conduct problems; hyperactivity/inattention; peer relationship problems; and prosocial behavior. The first four dimensions gathered generate a total difficulties score.

#### 2.2.3. Health Kids Resilience Assessment Module (HKRA)

To measure resilience and coping strategies, we used the Resilience Assessment Module from the California Healthy Kids Survey [26,31]. The California Healthy Kids Survey (CHKS), adapted for Portuguese students by Martins [32], is a survey of school climate and safety, student wellness, and youth resiliency. It comprises 58 items to evaluate students’ resilience traits—external and internal resources distributed by school, family, community, and peer group. It is a self-reported questionnaire that uses a 4-point Likert scale to assess each item.

### 2.3. Intervention

Data collection took place at two moments: a pre-test, before TM/QT learning, and a 3-month post-test. We used self-reported questionnaires which included a battery of instruments to evaluate the psychological dimensions in the study. The questionnaires were applied in groups, by the researchers, and by the coordinator of the implementation of the TM technique in the school context. Questionnaires were fulfilled in a classroom setting or at a specific space (time) chosen for that purpose. Standard instructions for all participants, read aloud, were used to ensure equal treatment. To guarantee the confidentiality of the data, questionnaire matching between pre-test and post-test was conducted using an alphanumeric code created by the participants themselves.

The sample of participants was constituted only of students who expressed an interest in learning and practicing transcendental meditation. However, students were split up into two groups: the TM group and the control group. This means that the control group learned the transcendental meditation technique with a three-month delay. This allowed us to ensure these students still benefitted from the technique while acting as a control group for the TM group (experimental group), which started immediately.

This study received approval by the Education, Audiovisual and Culture Executive Agency (approval code: 2017-3341/001-001) in January 2018.

### 2.4. Data Analysis

The Kolmogorov–Smirnov test was used to assess the normality of the distribution of all dependent variables. The Levene test was used to test for the homogeneity of variance of the dependent variables, before and after the intervention. All measures violated the normality assumption (*p* < 0.050). Homogeneity of variance was present for all variables (*p* > 0.050).

A repeated measures ANOVA was conducted to test for within-subject differences (before and after the intervention), between-subject differences (control and experimental groups), and the interaction of both factors (before and after, control and experimental).

## 3. Results

After the intervention’s twelve-week interval, the experimental group included 30 individuals who chose not to meditate; 38 who meditated 1 to 3 times a week; 9 who meditated 3 to 7 times a week; 8 who meditated 7 to 10 times a week; and 6 who meditated more than 10 times a week.

Both groups were similar at baseline. There were no significant differences in the means of response at baseline regarding anxiety, strengths and difficulties, and resilience (Table 1). It is possible to observe, in the results, a modification of the scores from the first to the second measure. The response means suggests a decrease in the dimensions of anxiety, and in the internalization and externalization of behavioral problems. There seemed to be an increase in prosocial behaviors (measured in reverse), and in resilience resources. There were no differences between groups, after the intervention, through the analysis of the comparison of means.

The repeated measures ANOVA (Table 2) allowed the observation of a within-subjects effect. Both groups presented a difference in humiliation, public performance fears, separation anxiety, anxious coping, tense/restless, and somatic/autonomic symptoms after the intervention. There was also a before and after intervention effect for internalizing problems and prosocial behavior. Both groups were less anxious and presented more strengths after the intervention period.

There were no between-subjects effects in this study.

An interaction effect was observed in the externalization of symptoms (Figure 1). The results suggest that there is an effect of the transcendental meditation program in controlling the externalization of difficulties such as the presence of hyperactive behaviors, which in this case, seem to reduce in the experimental group while increasing in the control group.

## 4. Discussion

The present study examined anxiety, resilience, and strengths and difficulties in students before and after TM practice, with a twelve-week interval, in both experimental and control groups. According to the previous literature, it was expected to observe an effect of meditation on psychological dimensions, favoring the experimental group. However, our study fails to attain those expectations.

Both groups were similar at the beginning point. After a twelve-week period, it was possible to observe that both groups had no observed mean differences in the psychological dimensions measured.

An interaction effect was observed on externalizing problems. The results suggest an effect of the QT/TM program in controlling the externalization of difficulties such as the presence of hyperactive behaviors, reducing in the experimental group while increasing in the control group. The effect of the QT/TM program seemed to be limited to the externalization of less adaptative behaviors. TM is described in the literature to have that outcome in behavior. Regular TM practice can neutralize the stresses and strains accrued during the day. Studies indicate that TM results in improvements in physical health (e.g., it reduces hypertension and cardiovascular disease), and psychological health (e.g., it reduces anxiety, depression, and anger), as well as influencing behavior outcomes, becoming more adaptative [33,34,35,36]. In our study, learning and practicing transcendental meditation seemed to be a resource to control the physical expression of anxiety and less positive emotional and mental states.

After a twelve-week period, a decrease in anxiety and in the internalization of behavior problems was observed for both groups. Additionally, an increase in prosocial behavior was also observed for both groups.

As previous studies with students and younger individuals, for example, the study of Valosek and colleagues [22], suggest that meditation is linked to positive emotional well-being and behavior, both groups seemed to change their behavior over time.

Our first hypothesis to explain these results concerns the expectation generated around the QT/TM program. Students were selected from those who volunteered, and the program was known within the whole school community; thus, some interaction may be found between students from the control and experimental groups. Additionally, meditation seems to have a positive evaluation and expectancy to reduce stress and anxiety, despite previous information that could be given or not. Woolfok and Rooney, in 1981 [37], observed that students maintained their beliefs about the impact of meditation on stress and anxiety, despite the initial information that was given to them. Both groups, given initial and contrary expectations about meditation, had similar positive effects on their self-assessment. They observed that despite the expectation’s manipulation, participants maintained their previous, positive evaluation of meditation to prevent stress. Although that study had different objectives and contributes to the positive and real impact of meditation in our lives, it does show that participants have positive expectations for mediation, and our students might have been tainted by the expectation of participating in the study, despite being in the control group and receiving no instructions on how or when to meditate.

A second possible explanation regards compliance with the meditation task and implementation. A two-times-a-day policy of meditation was expected to be followed. However, we had no control over meditation task implementation. Likewise, children’s attention and involvement in meditation tasks are different from other age groups, namely, adults. Studies concerning children and adolescents are still sparse, and an age limit is yet to be defined, although some authors may consider at least the stage of formal operations (12 yo) to attain full engagement with meditative and mindful techniques [38]. Our sample ranged from 9 to 16 years of age, which might present some difficulties in engaging and using the TM technique, although it is considered feasible for children to learn, and that might have influenced our results.

Additionally, another possible explanation should be taken into consideration. A meta-analysis developed to observe the effects of the transcendental meditation technique on trait anxiety [39] considered that the effect size of the TM technique on reducing trait anxiety depends on participants’ initial anxiety levels. Participants’ initial levels were not elevated. Therefore, this could be a possible explanation for our results where there was a decrease over time, which might be explained by contextual factors (e.g., evaluation term, work overload), but there were no differences between groups after the intervention. Both groups were less anxious and presented more strengths after the intervention period, probably because they were constrained by the same contextual issues.

Prior research on this type of program demonstrated a decrease in psychological distress, and increased coping in young adults at risk for hypertension. This mind–body program may reduce the risk for future development of hypertension in young adults [33,34]. Some studies also showed a change in resilience, associated with a change in emotional exhaustion, after controlling for baseline emotional exhaustion [20].

The study strengths include the randomized controlled design, with subjects allocated to either immediate start of meditation or wait-list control groups. The generalizability of the study results is limited to students who would be interested in participating in a meditation program.

However, the results suggest some useful guidelines for possible changes in practice in classroom contexts. The judgment made on students’ knowledge and skills is submitted to permanent personal comparison (internal) and social comparison processes (external). Accordingly, students compare their current performance with results previously achieved. They also compare it to peers’ accomplishments. Peers judge them equally. The acquisition of psychological resources that improve how they deal with errors, or the way they deal with the simple possibility of failure, tends to create favorable psychological functioning conditions. These improve students’ ability to face evaluative situations head-on, namely, in the context of the classroom and in the context of social interactions at school. This resource can improve students’ mental health and, consequently, create favorable conditions for when they must face situations that involve the risk of rejection, criticism, or humiliation, as it can be found in situations of evaluation of classroom performance.

Future research should investigate the effects on long-term effects of the school climate as well as psychological and behavioral outcomes of participants engaging in meditation. Concerning this last point, a better understanding of the extent to which behavioral and mental health changes depend on age or on a continued practice of transcendental meditation should be addressed in further studies. Another interesting issue could be the effect on the whole school climate, considering the number of students who practice versus those who do not practice TM. In the future, another type of intervention (e.g., yoga) could also be addressed to better understand if the effects are only related to the meditation, if they are influenced by the perceived different environment, or if they are influenced by having a quiet moment in daily activities.

The purpose of TM is that it can be applied without distinction in clinical and non-clinical populations.

## Figures and Tables

**Figure 1 children-08-00689-f001:**
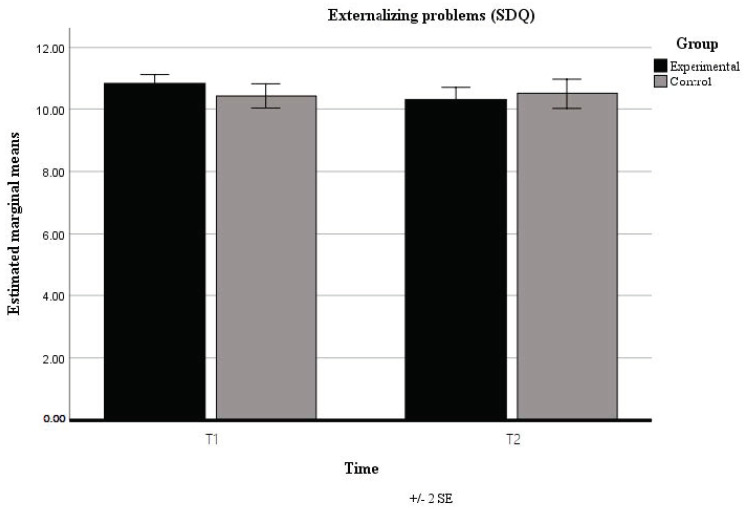
Interaction between time and group in externalizing problems (SDQ).

**Table 1 children-08-00689-t001:** Descriptive results of the repeated measures.

Variable	Before			After		
	Experimental	Control	Mean	Experimental	Control	Mean
	(Mean ± SD)		Differences (*p*)	(Mean ± SD)		Differences (*p*)
**Anxiety**						
Humiliation	5.88 ± 3.580	4.82 ± 3.358	0.057	3.92 ± 3.373	3.60 ± 2.775	0.521
Public performance fears	4.89 ± 2.334	4.68 ± 2.376	0.573	3.30 ± 2.170	3.22 ± 2.211	0.832
Separation anxiety	6.99 ± 2.882	7.23 ± 3.561	0.640	5.02 ± 2.793	5.57 ± 3.283	0.435
Perfectionism	6.98 ± 1.763	7.01 ± 2.098	0.934	7.25 ± 0.121	6.90 ± 1.996	0.287
Anxious coping	7.83 ± 2.559	7.82 ± 2.465	0.999	6.78 ± 2.743	6.44 ± 3.001	0.451
Tense/Restless	4.18 ± 2.931	3.91 ± 2.650	0.532	3.14 ± 2.588	3.05 ± 2.184	0.814
Somatic/Autonomic	3.16 ± 2.623	3.23 ± 2.852	0.879	2.02 ± 1.960	2.08 ± 2.442	0.856
**Strengths and Difficulties**						
Internalizing problems	9.03 ± 2.079	9.17 ± 1.896	0.661	8.89 ± 1.977	8.51 ± 1.916	0.211
Externalizing problems	10.82 ± 1.572	10.42 ± 1.579	0.107	10.30 ± 2.059	10.50 ± 1.818	0.523
Prosocial behavior	8.40 ± 1.081	8.45 ± 1.180	0.786	7.88 ± 1.481	7.99 ± 1.359	0.638
**Resilience**						
External resources	3.09 ± 0.416	3.06 ± 0.487	0.613	3.13 ± 0.503	3.12 ± 0.499	0.954
Internal resources	3.19 ± 0.459	3.13 ± 0.567	0.429	3.22 ± 0.606	3.15 ± 0.653	0.497
Response set breakers	2.96 ± 0.359	2.94 ± 0.475	0.782	3.02 ± 0.474	2.98 ± 0.514	0.585

**Table 2 children-08-00689-t002:** ANOVA with repeated measures.

	Time			Group			Time * Group		
	*F*	*p*	*η* ^2^ *_p_*	*F*	*p*	*η* ^2^ *_p_*	*F*	*p*	*η* ^2^ *_p_*
**Anxiety**									
Humiliation	49.299	<0.001 *	0.229	2.113	0.148	0.016	2.624	0.107	0.013
Public performance fears	72.773	<0.001 *	0.305	0.208	0.649	0.001	0.145	0.704	0.0008
Separation anxiety	50.371	<0.001 *	0.233	0.515	0.474	0.003	0.075	0.784	0.0004
Perfectionism	0.216	0.643	0.001	0.353	0.553	0.002	1.393	0.240	0.008
Anxious coping	29.106	<0.001 *	0.149	0.224	0.637	0.001	0.565	0.453	0.003
Tense/Restless	17.998	<0.001 *	0.098	0.277	0.599	0.002	0.179	0.673	0.001
Somatic/Autonomic	51.096	<0.001 *	0.235	0.001	0.992	0.0001	0.033	0.857	0.0006
**Strengths and Difficulties**									
Internalizing problems	7.046	0.009 *	0.041	0.204	0.652	0.001	3.069	0.082	0.018
Externalizing problems	2.177	0.142	0.013	0.181	0.671	0.001	4.237	0.041 *	0.025
Prosocial behavior	20.988	<0.001 *	0.112	0.202	0.654	0.001	0.074	0.786	0.0004
**Resilience**									
External resources	2.245	0.136	0.013	0.091	0.764	0.0005	0.209	0.648	0.001
Internal resources	0.250	0.618	0.002	0.700	0.404	0.004	0.002	0.963	0.0001
Response set breakers	2.115	0.148	0.013	0.243	0.622	0.001	0.112	0.738	0.0007

* stands for significant results.

## Data Availability

Not applicable.

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
