# Peer review of "Meditation Effects on Anxiety and Resilience of Preadolescents and Adolescents: A Randomized Controlled Study"

_children, 2021, doi:10.3390/children8080689_

Round 1

Reviewer 1 Report

This is a very well-written report on the meditation effects on axiety and resilience of adolescents. The analyses conducted are clearly explained and the contribution to research is properly elaborated. The paper make a significant contribution to the research which have the potential to inform practice in important ways. Major recommendations are suggested in order to enhance the usefulness of this manuscript.

Firstly, I strongly suggest to provide details about the process of obtaining consent to participate in this study.

- Please, inculde the number of participants who were approached.

-  the number of patients who refused to participate as well as missing data.

- how were participants selected?

- There were some inclusión/exlusion criteria?

- Sample size power estimation should be included given the nature of the intervention.

-Why authors selected only a school?

Additionally, authors should included the code of the ethic committe and if available the number of register in a repository.

In the results section authors should include the effect size in all analyses. It would be great if sociodemographic data showing no signficant baseline differences between groups could display in table.

Last, how educators should take into account the present findings in the routine of classroom? And how results fit with future intervention approach within school settings?

Author Response 

Reviewer 1

Reviewer Comments

Authors Answers

This is a very well-written report on the meditation effects on axiety and resilience of adolescents. The analyses conducted are clearly explained and the contribution to research is properly elaborated. The paper make a significant contribution to the research which have the potential to inform practice in important ways. Major recommendations are suggested in order to enhance the usefulness of this manuscript.

Thank you so much for the general comment of our study.

Firstly, I strongly suggest providing details about the process of obtaining consent to participate in this study.

Information was clarified in the document:

“A full presentation of the program was made to school community: teachers and staff, students, and parents. An informed consent was delivered to all parents or legal equivalents. Teachers and staff gave their informed consent to participate and were also formed in the TM technique”.

- Please, include the number of participants who were approached.

Information was added:

“From the 300 consents received, 200 students were randomly selected to participate, but only 168 completed the full experimental design”.

-  the number of patients who refused to participate as well as missing data.

There weren´t participants who refused but some haven´t fulfilled the consent, and we have some experimental mortality between moments.

- how were participants selected?

Information was added:

“66 participants integrated the control group. Distribution was made considering the student’s interest in learning TM technique”.

- There were some inclusión/exlusion criteria?

Inclusion criteria: Fulfill the consent, learn the technique, and answer the questionnaires in both moments.

-Why authors selected only a school?

For several reasons: 1) the proximity between investigation and implementation teams; 2) the number of students interest in the technique.

Additionally, authors should included the code of the ethic committe and if available the number of register in a repository.

Information added:

Education, Audiovisual and Culture Executive Agency

Approval code: 2017-3341/001-001

Date: 15/01/2018Education, Audiovisual and Culture Executive Agency

Approval code: 2017-3341/001-001

Date: 15/01/2018

In the results section authors should include the effect size in all analyses. It would be great if sociodemographic data showing no signficant baseline differences between groups could display in table.

There is information about in in the sample description:

“There were no differences at baseline considering age (p=.714), sex (p=.864), school year (p=.406). Having a retention is associated to the experimental group (χ2=4.947, p=.026).”

Last, how educators should take into account the present findings in the routine of classroom? And how results fit with future intervention approach within school settings?

Information was clarified in the discussion.

The judgment made to students' knowledge and skills is submitted to permanent personal comparison (internal) and social comparison processes (external). Respectively, students compare their current performance with results previously achieved. They also compare it to peers' accomplishments. Peers judge him equally. The acquisition of psychological resources that improve how they deal with errors, or the way they deal with the simple possibility of failure, tend to create favorable psychological functioning conditions. These improve students' ability to face evaluative situations head-on, namely, in the context of the classroom and in the context of social interactions at school. This resource can improve the students' mental health and, consequently, create favorable conditions for when they must face situations that involve the risk of rejection, criticism, or humiliation, as situations of evaluation of classroom performance can be.

Reviewer 2 Report

This paper is interesting because it applies a new approach of meditation technique in preadolescents and adolescents measuring the possible effects on anxiety, psychological difficulties and resilience. The frequency of participants is good, however, there are some problems with the experimental design and some clarifications are needed.

In the title you put the term adolescents but then you didn’t take into consideration only them. Change the title putting preadolescents and adolescents.

Insert the response rate and the reasons for why some adolescents refused the participation. This element should be cited in the discussion section, because it is probably related to the fact that the participants are all interested to this meditation approach. Did they act this approach yet? Probably this element should be asked to them.

It is not clear how the assignment of the experimental group was made. It should be adopted a randomized methodology for the assignment. You wrote that the “Distribution was made considering the student’s interest in learning TM technique”, what you intend and why did you use this criterion?

Why the frequency of participants was different along experimental and control group?

Probably the control group should follow another intervention typology, the absence of intervention doesn’t give value to this type of meditation intervention, but only tell the difference between the existence of an intervention or not. You specified that after the intervention was run also to that of the control group, but there is not clarification on the control group procedure.

Why you didn’t run gender or ager differences in the resilience or wellbeing factors?

In the MASC there is a cut off scores? In the SDQ there are cut off scores: did you take into consideration these in the results?

Taking into consideration these aspects related to the results in the discussion and suggest also recommendations for future research in the field.

Author Response

Reviewer 2

Reviewer Comments

Authors Answers

This paper is interesting because it applies a new approach of meditation technique in preadolescents and adolescents measuring the possible effects on anxiety, psychological difficulties and resilience. The frequency of participants is good, however, there are some problems with the experimental design and some clarifications are needed.

In the title you put the term adolescents but then you didn’t take into consideration only them. Change the title putting preadolescents and adolescents.

The title was changed, considering the suggestion.

Insert the response rate and the reasons for why some adolescents refused the participation. This element should be cited in the discussion section, because it is probably related to the fact that the participants are all interested to this meditation approach. Did they act this approach yet? Probably this element should be asked to them.

Information added in the discussion section

It is not clear how the assignment of the experimental group was made. It should be adopted a randomized methodology for the assignment. You wrote that the “Distribution was made considering the student’s interest in learning TM technique”, what you intend and why did you use this criterion?

It was clarified. We agree, it was not properly explained.

Why the frequency of participants was different along experimental and control group?

Depends on their answer to the questionnaire, in a longitudinal study some mortality of the samples occurs.

Probably the control group should follow another intervention typology, the absence of intervention doesn’t give value to this type of meditation intervention, but only tell the difference between the existence of an intervention or not. You specified that after the intervention was run also to that of the control group, but there is not clarification on the control group procedure.

In fact, for a new study those kind of reflections will be considered, also we mentioned in the discussion section.

Why you didn’t run gender or ager differences in the resilience or wellbeing factors?

The age and gender variables are distributed similarly in the control and experimental groups. Any existing differences will be similar in both groups. In fact, it is expected that they exist (with age, resilience, and strategies to deal with anxiety increase, for example), but the existing differences should be evenly distributed between groups. These do not differ in the 1st moment. In fact, these differences don't have much relief, showing effect sizes between small and medium.

In the MASC there is a cut off scores? In the SDQ there are cut off scores: did you take into consideration these in the results?

The purpose of TM is that it can be applied without distinction in clinical and non-clinical populations. In this case, we were interested in possible variations in measurements with some sensitivity in a normal population, but without a clinical focus. MASC indicates the presence of anxious symptoms, without cutoff values, being more useful to identify the type or expression of anxiety than to determine clinically anxious patients.

The SDQ presents cutoff values for interpreting its results, however, the perspective was not of intervention in a clinical sample. If all children have their growth challenges, the SDQ is sensitive enough to show variations in their expression, without being in a clinical or behavioral disturbance dimension.

Taking into consideration these aspects related to the results in the discussion and suggest also recommendations for future research in the field.

Some required information was added in the discussion section.

Round 2

Reviewer 2 Report

I found the paper really ameliorated. I have no more considerations to do. 

Compliments for your hard work!